# Putative Mechanisms Responsible for the Antihyperglycemic Action of *Lactobacillus paracasei* HII01 in Experimental Type 2 Diabetic Rats

**DOI:** 10.3390/nu12103015

**Published:** 2020-10-01

**Authors:** Parichart Toejing, Nuntawat Khat-Udomkiri, Jannarong Intakhad, Sasithorn Sirilun, Chaiyavat Chaiyasut, Narissara Lailerd

**Affiliations:** 1Department of Physiology, Faculty of Medicine, Chiang Mai University, Chiang Mai 50200, Thailand; lookplanoi_@hotmail.com (P.T.); Jannarong.7051@gmail.com (J.I.); 2Innovation Center for Holistic Health, Nutraceuticals and Cosmeceuticals, Department of Pharmaceutical Sciences, Faculty of Pharmacy, Chiang Mai University, Chiang Mai 50200, Thailand; mysan_t_u_s@hotmail.com (N.K.-U.); sasithorn.s@cmu.ac.th (S.S.); chaiyavat@gmail.com (C.C.)

**Keywords:** type 2 diabetes mellitus, gut microbiota, *Lactobacillus paracasei*, antihyperglycemia

## Abstract

Despite the updated knowledge of the impact of gut dysbiosis on diabetes, investigations into the beneficial effects of individual bacteria are still required. This study evaluates the antihyperglycemic efficacy of *Lactobacillus paracasei* HII01 and its possible mechanisms in diabetic rats. Diabetic rats were assigned to receive vehicle, *L. paracasei* HII01 (10^8^ CFU/day), metformin 30 (mg/kg) or a combination of *L. paracasei* HII01 and metformin. Normal rats given vehicle and *L. paracasei* HII01 were included. Metabolic parameters, including in vitro hemi-diaphragm glucose uptake, skeletal insulin-signaling proteins, plasma lipopolysaccharide (LPS), gut permeability, composition of gut microbiota and its metabolites, as well as short-chain fatty acids (SCFAs), were assessed after 12 weeks of experiment. The results clearly demonstrated that *L. paracasei* HII01 improved glycemic parameters, glucose uptake, insulin-signaling proteins including pAkt^Ser473^, glucose transporter 4 (GLUT4) and phosphorylation of AMP-activated protein kinase (pAMPK^Thr172^), tumor necrosis factor (TNF-α) and nuclear factor-κB (NF-kB) in diabetic rats. Modulation of gut microbiota was found together with improvement in leaky gut, endotoxemia and SCFAs in diabetic rats administered *L. paracasei* HII01. In conclusion, *L. paracasei* HII01 alleviated hyperglycemia in diabetic rats primarily by modulating gut microbiota along with lessening leaky gut, leading to improvement in endotoxemia and inflammation-disturbed insulin signaling, which was mediated partly by PI3K/Akt signaling and AMPK activation.

## 1. Introduction

Type 2 diabetes mellitus (T2DM), a multifactorial metabolic endocrine disorder, is characterized by persistent hyperglycemia, and it is basically a result of insulin resistance and impaired β-cell function. According to the International Diabetes Federation (IDF), the number of diabetic patients worldwide was 425 million in 2017 and will rise to 629 million by 2045 [1]. Although, several influences such as genetics, age, unhealthy lifestyle and obesity are accepted as risk factors of T2DM [2]. Nowadays, it is well accepted that gut microbiota is linked to the development of T2DM [3]. Changes in gut microbiota composition, known as gut dysbiosis, have been associated with disrupted gut barrier functions and increased gut permeability [4,5]. The enhancement of gut permeability might result in bacterial lipopolysaccharide (LPS) leak into blood circulation, followed by inflammatory activation through the LPS-Toll-like receptor 4-Nuclear factor-κB (LPS-TLR4-NF-kB) signaling pathway [6,7]. Moreover, tumor necrosis factor (TNF-α), a pro-inflammatory cytokine that is generated from the LPS-TLR4-NF-kB signaling pathway, induces the enhancement of the phosphorylation of insulin receptor substrate 1 (IRS-1^Ser307^) [8]. The serine phosphorylation of IRS-1 blunts the activation of the phosphatidylinositol 3-kinase (PI3K)/protein kinase B (Akt) signaling pathway, resulting in a reduction in glucose transporter 4 (GLUT4) translocation and glucose uptake in the skeletal muscle, which causes insulin resistance and hyperglycemia [9]. Therefore, the modulation of gut microbiota is used as a strategy for prevention or adjuvant treatment of T2DM. 

Probiotics are defined as “live microorganisms which when administered in adequate amounts confer a specific health benefit on the host” [7]. Conclusive evidence indicates that modulation of gut microbiota by probiotics provides beneficial health effects in both animal and clinical research of T2DM [10,11,12]. Among probiotics, Lactobacillus is one of the most popular strains that have been used for investigation [13]. The oral administration of *Lactobacillus reuteri* GMNL-263 decreased the plasma glucose level in high fructose-fed rats [14]. Lim et al., 2016, also revealed that gut tight junction, endotoxemia and inflammation were ameliorated after *Lactobacillus sakei* OK67 treatment in type 2 diabetic rat model [15]. Furthermore, a previous study demonstrated that the production of short-chain fatty acids (SCFAs) and gut microbial metabolites, including acetate, propionate and butyrate, seems to play an important role in the attenuation of T2DM [16].

Recently, a newly identified probiotic strain *Lactobacillus paracasei* spp. HII01, from the fermentation of northern Thai pickle, showed a significant improvement in gut dysbiosis and metabolic endotoxemia in obese rats [17]. In addition, *L. paracasei* HII01 restored kidney function by attenuating insulin resistance and hyperglycemia in obese rats [18]. However, no information is available on the antidiabetic potential of *L. paracasei* HII01. Therefore, this study was conducted to evaluate the antidiabetic effect of *L. paracasei* HII01 on experimental type 2 diabetic rats and explore the possible underlying mechanisms.

## 2. Materials and Methods

### 2.1. Animals and Ethical Approval

Adult male Wistar rats weighing approximately 180–200 g were used in this study. All rats were obtained from the National Laboratory Animal Center, Mahidol University, Thailand. The experimental protocol was approved by the Research Animal Care and Use Ethical Committee, Faculty of Pharmacy, Chiang Mai University, Thailand (Ethics approval no. 04/2015). All animals were housed under controlled temperature at 25 ± 2 °C with a 12 h light/dark cycle and were fed with a standard rodent chow diet and water ad libitum. The animals were given an acclimatization period of 1 week. The animals used in this study were cared for according to the principles and guidance of the “Guide for the Care and Use of Animals in compliance with the National Institute of Health Guideline for the Care and Treatment of Animals”. 

### 2.2. Stock and Cultivation of the Strain

Lactobacillus strain No. HII01 is a novel non-human origin-isolated strain of lactic acid-producing bacteria that has been approved by the Food and Drug Administration (FDA), Thailand. It was prepared at the Innovation Center for Holistic Health, Nutraceuticals and Cosmeceuticals, Faculty of Pharmacy, Chiang Mai University. The 16S rRNA gene sequence of the representative strain showed 99.0% similarity, 1511 bps, to *L. paracasei* accession number AP012541.1. The bacterial strain was revived in MRS (de Mann Rogosa Sharpe) (Difco Detroit, MI, USA) broth with pH of 6.5 + 0.2 at 25 °C. The stock culture of the HII01 was maintained at 20% (*v*/*v*) glycerol-MRS broth at −70 °C. The organism was activated 3 times in MRS broth using 1% (*v*/*v*) inoculum at 37 °C for 24 h until further use. 

### 2.3. Bacterial Culture

The growth culture of the strain (1%) was inoculated into freshly prepared MRS. The bacterial cell of HII01 was prepared from the late exponential growth phase of cell growth. The inoculum of the strain in the culture medium was collected by centrifugation at 10,000× *g*, 4 °C for 10 min. The supernatant was discarded, and the cell pellet was washed 3 times with phosphate buffer saline (pH 7.0 ± 0.2). Then, the cell pellet was re-suspended, and a final concentration of approximately 10^8^ colony forming unit (CFU)/mL sterile distilled water was used in the experiment. 

### 2.4. Induction of Experimental Diabetes

The establishment of a type 2 diabetic model was carried out as described by Srinivasan et al., 2005 [19]. The rats were assigned into two dietary regimens by feeding them with standard rodent chow diet (10.95% kcal energy from fat source) or high-fat diet (53.63% kcal energy from fat source) (Appendix A) ad libitum. After 2 weeks of initial dietary period, diabetes mellitus was induced in overnight fasted rats with a single intraperitoneal injection of streptozotocin (STZ) (Sigma-Aldrich, St. Louis, MO, USA) dissolved in citrate buffer (pH 4.5) at a dose of 40 mg/kg. After 14 days of induction, diabetes mellitus was confirmed by the fasting plasma glucose levels. The rats with fasting plasma glucose level ≥250 mg/dL without hypoinsulinemia were considered to exhibit type 2 diabetes and were included in this study. A total of 60 male Wistar rats were randomly divided into six groups (n = 10 per group): normal diet control (NDC), normal rat supplemented with *L. paracasei* HII01 (10^8^ CFU/day) (ND-L), diabetic rat control (DMC), diabetic rat supplemented with *L. paracasei* HII01 (10^8^ CFU/day) (DM-L), diabetic rat treated with metformin (30 mg/kg) (DMM) as the positive control and diabetic rat supplemented with a combination of *L. paracasei* HII01 (10^8^ CFU/day) and metformin (30 mg/kg) (DMM-L). After 12 weeks of supplementation, overnight fasted rats were sacrificed via an intraperitoneal injection of overdose Nembutal^®^ (Liboume, France). Blood samples were collected in appropriate anticoagulant and then centrifuged at 13,000× *g* for 1 min to obtain plasma. The soleus muscle, gastrocnemius muscle and liver were rapidly removed, frozen in liquid nitrogen and stored at −80 °C for further analysis.

### 2.5. Biochemical Analysis of Plasma

The plasma levels of glucose, triglyceride (TG), cholesterol, low-density lipoprotein cholesterol (LDL) and high-density lipoprotein cholesterol (HDL) were analyzed using a commercial kit (Biotech, Bangkok, Thailand). The plasma insulin, leptin and adiponectin levels were measured using a rat ELISA kit (LINCO Research, Charles, MO, USA) following the instructions of the manufacturer. The degree of insulin resistance was assessed by the homeostasis model assessment of insulin resistance (HOMA-IR), calculated from fasting plasma insulin and glucose concentrations [20]. The HOMA-IR index was calculated using the following formula: HOMA-IR = [fasting plasma insulin level (ng/dL) × fasting plasma glucose level (mg/dL)]/405.1

### 2.6. Oral Glucose Tolerance Test

Oral glucose tolerance test (OGTT) was performed on the 11th week. All rats were fasted overnight and the fasting plasma glucose was collected prior to glucose administration (time = 0) as the baseline value. Then, 2 g/kg of glucose solution was administered by oral gavage. The blood samples were collected at 15, 30, 60 and 120 min after glucose administration. The plasma glucose levels were determined, and the area under the curve (AUC) for glucose was calculated to assess glucose tolerance using the trapezoidal rule [21].

### 2.7. In Vitro Glucose Uptake by Isolated Rat Hemi-Diaphragm

Glucose uptake by isolated hemi-diaphragm was determined according to the methods described by Thabet et al., 2008, with some modifications [22]. The glucose uptake was divided into 2 experimental conditions, including without and with insulin (0.25 IU/mL) to determine the basal and insulin-stimulated glucose uptake, respectively. After overnight fasting, the rats were sacrificed with intraperitoneal injection of overdose Nembutal^®^. The diaphragm of the rat was rapidly removed with minimal trauma, divided into two halves and rinsed in cold balanced salt solution (BSS) to remove any blood clot. Each hemi-diaphragm was placed in a conical flask containing 3 mL of BSS and incubated with carbogen (95% O_2_/5% CO_2_) with shaking at 100 cycles/min for 90 min at 37 °C. At the end of the incubation period, the isolated hemi-diaphragm was removed, blotted with filter paper and weighed. An aliquot of the incubation medium was used for measurement of glucose concentration. Glucose uptake per gram of tissue was calculated as the difference between the initial and final glucose content in the incubated medium. 

### 2.8. In Vivo Intestinal Permeability Assay

Gut permeability was assessed at the end of the experiment. This assay is an indirect measure of total intestinal permeability. The principle of this assessment is based on the intestinal leakage of 4000 Da Fluorescein isothiocyanate–dextran (FITC–dextran) into blood circulation. Briefly, rats were fasted overnight, and blood samples were collected as the negative control of the experiment to determine the background of rat plasma. Then, FITC–dextran (600 mg/kg) (Sigma-Aldrich, St. Louis, MO, USA) was administered to the rats by oral gavage, and blood samples were collected at 2.5 and 5 h later. The blood sample was immediately centrifuged at 6000 rpm for plasma separation, and the plasma was diluted with an equal volume of phosphate buffered saline (PBS) (pH 7.4). The plasma concentration of the FITC–dextran was determined using a Synergy^™^ H4 fluorescene microplate reader (BIOTEK^®^ Instruments, Inc., Vermount, VT, USA) with an excitation wavelength of 485 nm and an emission wavelength of 535 nm compared with the standard curve of serially diluted FITC–dextran [23,24].

### 2.9. Determination of Plasma Lipopolysaccharide (LPS)

The plasma LPS level was determined using QCL-1000TM Endpoint Chromogenic Limulus Amebocyte Lysate (LAL) Assay Kit (Lonza, Verviers, Belgium) following the instructions of the manufacturer. Briefly, plasma was mixed with LAL reagent and incubated at 37 °C in a heating block for 10 min, followed by the addition of substrate solution and final incubation at 37 °C for 6 min. After that, the stop reagent was added. The presence of LPS in the plasma was inferred by the development of yellow color. The absorbance of the sample was quantified using spectrophotometry at 405–410 nm [25].

### 2.10. Determination of Triglyceride Accumulation in Liver and Skeletal Muscle

The liver and gastrocnemius muscle TG contents were measured according to the method of Frayn and Maycock, 1980, with slight modifications [26]. Briefly, a 0.05–0.2 g portion of the liver and muscle was minced and put into a glass tube containing 3 mL of chloroform-isopropanol 2:3 (*v*/*v*). The homogenate was pipetted into a glass tube and evaporated to dryness at 40 °C for 16 h. The dried residue was dissolved and mixed in 10% bovine serum albumin (BSA). The triglyceride contents were measured using a commercial colorimetric kit (Biotech, Bangkok, Thailand).

### 2.11. DNA Extraction from Fecal Samples

Bacterial DNA was collected from fecal samples (60–70 g) using NucleoSpin^®^ DNA stool kit (Macherey-Nagel, Dueren, Germany). All procedures were performed according to the manufacturer’s instructions. Qualitative analysis of bacterial DNA was evaluated by SPECTROstar Nano Absorbance microplate reader (BMG Labtech, Ortenberg, Germany). The ratio of the absorbance at 260 and 280 nm (A260/280) was used to identify the purity of nucleic acid specimen. An A260/280 value greater than 1.8 indicated a pure DNA sample. Bacterial DNA contents were evaluated using the relationship that 50 μg/mL of pure DNA sample represented an A260 of 1. 

### 2.12. q-PCR Assay Conditions and Cycle Threshold

Quantitative PCR (qPCR) analyses were carried out in 96-well optical plates on the Quantstudio TM6 Flex Real Time PCR System (Applied Biosciences, Warrington, U.K.). The amplification reaction was performed in a total of 20 μL containing 10 μL of SYBR^™^ master mix, 2 μL of fecal bacterial DNA sample, 1 μL of reverse primer, 1 μL of forward primer and 6 μL of deionized water. The group-specific primers of bacterial targets based on 16S rDNA sequences are listed in Appendix A. qPCR was conducted as follows: Uracil-DNA Glycosylase (UDG) activation step at 50 °C for 2 min followed by initial denaturation at 95 °C for 2 min and 40 cycles of denaturation step at 95 °C for 20 s and the annealing/extension step at 60 °C for 20 s. Melt curve analysis was then performed after each run to check the non-specific amplification of the primers. The cycle threshold (Ct) of bacterial DNA was calculated by absolute quantification strategy using the standard curve of the target bacterial strain. The result was expressed as log CFU/mL.

### 2.13. Measurement of Organic Acid Contents in Cecal Samples

The amounts of organic acids (acetic, propionic, butyric and lactic acids) in cecal content and fecal samples were measured by high-performance liquid chromatography (HPLC), as described previously [27]. Briefly, the sample was homogenized in 0.15 mM sulfuric acid and centrifuged at 10,000× *g* at 4 °C for 10 min. The supernatant was collected and filtered through 0.22 μm nylon syringe filter. The samples were analyzed by a Shimadzu HPLC system using Shodex SUGAR SH1011 (SHOWA DENKO K.K., Tokyo, Japan). The detection was carried out using a UV detector at 210 nm, and the column temperature was maintained at 75 °C. The samples were isocratically eluted with 5 mM sulfuric acid at 0.6 mL/min. The concentration of organic acids was quantified by comparison with the standard curve, and the results were expressed as μmol/g sample.

### 2.14. Western Blot Analysis

The soleus muscle of the rat was obtained after sacrifice. The homogenates were centrifuged at 4 °C for 10 min at 10,000× *g*, and the supernatants were used for Western blot. Total protein concentration was determined using a Bradford protein assay kit (Bio-Rad Laboratories, Hercules, CA, USA). Then, 30–50 ug of proteins was loaded in 10% sodium dodecyl sulphate–polyacrylamide gel electrophoresis (SDS-PAGE) for protein separation. The proteins were transferred to nitrocellulose membrane and blocked with blocking buffer for 1 h at room temperature with gentle shaking, followed by incubation overnight at 4 °C with specific primary antibody, Akt (Millipore Corporation, Burlington, MA, USA), phosphorylation of pAkt^Ser473^ (Millipore Corporation, Burlington, MA, USA), AMP-activated protein kinase-α (AMPK-α) (Millipore Corporation, Burlington, MA, USA), pAMPKα^Thr172^ (Millipore Corporation, Burlington, MA, USA), GLUT4 (Chemicon International, Temecula, USA), TNF-α (Millipore Corporation, Burlington, MA, USA) and NF-kB (Santa Cruz biotechnology, Dallas, TX, USA). After incubation with the primary antibody, the membrane was washed and incubated with horseradish peroxidase-conjugated secondary antibody for 1 h at room temperature and rewashed again. The protein bands in the membranes were identified by enhanced chemiluminescence (ECL) detection reagent (GE Healthcare, Piscataway, NJ, USA). The concentration of protein was expressed by comparison with the mean value in the NDC group, which was arbitrarily set as 100.

### 2.15. Statistical Analysis

The results are presented as the mean value ± standard error of the mean (SEM). To detect the effects of treatment on the blood and fecal parameters among the six experimental groups, one-way analysis of variance (ANOVA) followed by Least-Significant Different (LSD) post-hoc analysis was used to determine significant differences between groups. The SPSS Advanced Statistics software (version17 SPSS Inc., Chicago, IL, USA) was used for statistical analysis. In all cases, a *p*-value less than 0.05 was used and considered to be statistically significant.

## 3. Results

### 3.1. Effects of L. paracasei HII01 on Body Weight (BW), Visceral Fat (VF) Weight and Visceral Fat/100 g BW

The BW, VF weight and VF/100 g BW of all experimental groups are represented in Table 1. The initial body weight was very similar in all experimental groups (394.5 ± 7.47 g, 389.00 ± 7.02 g, 389.5 ± 5.98 g, 386.5 ± 6.41 g, 383.75 ± 7.78 g and 393.50 ± 8.56 g, respectively). Following 12 weeks of oral administration of L. paracasei HII01, the BW, VF weight and VF/100 g BW did not differ among the normal rats. However, the DMC group had a significant increase in BW, VF weight and VF/100 g BW compared with the NDC group (*p* < 0.05), which indicates visceral obesity. Interestingly, the BW, VF weight and VF/100 g BW of the DM-L group were significantly decreased compared with the DMC group (*p* < 0.05). Likewise, the DMM and DMM-L groups had significantly lower values of the mentioned variables compared with the DMC group (*p* < 0.05). The above findings were observed in the absence of significant alterations in the food intake among the diabetic groups (93.86 ± 4.88 g/day, 90.00 ± 4.59 g/day, 89.43 ± 3.74 g/day and 97.00 ± 3.95 g/day, respectively). 

### 3.2. Effects of L. paracasei HII01 on Glycemic Control and Plasma Adipokine Hormones

To explore the anti-hyperglycemic effect of *L. paracasei* HII01, the plasma biochemical parameters involved in glycemic control were measured at the end of the study. As shown in Figure 1, *L. paracasei* HII01 administration did not alter the fasting plasma glucose, insulin, leptin and adiponectin levels among the normal rats. Similarly, the oral administration of *L. paracasei* HII01 did not affect the HOMA-IR, a method used to quantify insulin resistance, in the normal rats (*p* > 0.05) (Figure 1C). These results established that the administration of *L. paracasei* HII01 in normal rats had no effect on glycemic parameters. The diabetic rats showed higher fasting plasma glucose and insulin levels as well as HOMA-IR compared with normal rats (*p* < 0.05). Remarkably, the administration of *L. paracasei* HII01, metformin alone or in combination with *L. paracasei* HII01 significantly ameliorated the fasting plasma glucose (−42.87%, −49.13% and −49.29%, respectively, *p* < 0.05) and insulin levels compared with the DMC group (−28.80%, −27.46% and −41.44%, respectively, *p* < 0.05). In accordance with these results, the HOMA-IR of the DM-L, DMM and DMM-L groups were significantly reduced compared with the DMC group (−59.38%, −62.54% and −69.29%, respectively, *p* < 0.05). 

Leptin and adiponectin are two adipokine hormones that play a crucial role in metabolic regulation and are involved in insulin sensitivity. Therefore, we assessed the plasma leptin and adiponectin levels. As illustrated in Figure 1D,E, there were no significant differences in the plasma leptin and adiponectin levels between the two normal experimental groups, while the DMC group showed significantly increased plasma leptin level compared with the normal rats, indicating that leptin resistance was developed in diabetes (150.46%, *p* < 0.05). The plasma leptin level significantly dropped in the DM-L, DMM and DMM-L groups compared with the DMC group (−41.58%, −39.33% and −36.24%, respectively, *p* < 0.05). However, the plasma adiponectin level in the DMC group significantly decreased compared with the NDC group (−21.35%, *p* < 0.05). The administration of *L. paracasei* HII01, metformin, as well as the combination of *L. paracasei* HII01 and metformin, significantly increased the plasma adiponectin level (25.37%, 26.31% and 25.85%, respectively, *p* < 0.05). 

### 3.3. Effects of L. paracasei HII01 on the Glucose Tolerance Test

To determine whether the administration of *L. paracasei* HII01 could affect the whole-body insulin sensitivity in type 2 diabetic rats, the OGTT was conducted on the rats after 11 weeks of intervention. As shown in Figure 2A,B, there were no significant differences in the plasma glucose levels at all time points and the AUC for glucose between the NDC and ND-L groups. As expected, the plasma glucose levels after glucose loading revealed significantly higher values in the DMC group at all time points compared with the NDC group (Figure 2A, *p* < 0.05). Compared with the NDC group, the incremental area under the curve (IAUC) was markedly increased in the DMC group (Figure 2B, *p* < 0.05). These findings proved that impaired glucose tolerance was established in T2DM rats. Notably, the glucose levels at all time points in rats supplemented with *L. paracasei* HII01, metformin alone or in combination with *L. paracasei* HII01 were significantly reduced in comparison with the DMC group (*p* < 0.05). There was significant reduction in the total area under the curve (TAUC) and IAUC values in the DM-L, DMM and DMM-L groups compared with the DMC group (*p* < 0.05). 

### 3.4. Effects of L. paracasei HII01 on Lipid Parameters

In the present study, we also examined the hypolipidemic effect of probiotic *L. paracasei* HII01 on type 2 diabetic rats. As revealed in Table 2, the levels of plasma TG, total cholesterol and LDL in the DMC group were significantly increased compared with the normal control rats at the end of the study (*p* < 0.05). Oral administration of *L. paracasei* HII01 or in combination with metformin significantly restored the plasma TG, total cholesterol and LDL levels compared with the DMC group (*p* < 0.05), while changes in the plasma TG, cholesterol and LDL levels were not observed in normal rats treated with *L. paracasei* HII01. However, no significant change in the plasma HDL level was displayed in the DMC group compared with the NDC group. All intervention groups had significantly increased plasma HDL levels compared with the NDC group (*p* < 0.05). 

### 3.5. Effects of L. paracasei HII01 on Tissue Triglyceride Accumulation

Next, we evaluated the effects of *L. paracasei* HII01 on TG accumulation in both the skeletal muscle and liver because the accumulation of lipid within target tissues of insulin is closely associated with insulin resistance and abnormal lipid metabolism. The muscle TG accumulation of the DMC group significantly increased compared with the NDC group (*p* < 0.05), as shown in Appendix A. A significant reduction in muscle TG accumulation was found in diabetic rats administered *L. paracasei* HII01, metformin alone or in combination with *L. paracasei* HII01 compared with the DMC group (−36.40%, −38.31% and −46.61%, respectively, *p* < 0.05). For the liver, the DMC group also demonstrated a significant increase in hepatic TG accumulation compared with the NDC group (*p* < 0.05). Interestingly, the administration of *L. paracasei* HII01 significantly reduced the hepatic TG accumulation compared with the DMC group (−21.65%, *p* < 0.05). The hepatic TG accumulation of the DMM and DMM-L groups tended to decrease compared with the DMC group (−14.12% and −10.95%, respectively, *p* > 0.05).

### 3.6. Effects of L. paracasei HII01 on In Vitro Skeletal Muscle Glucose Uptake

To examine whether *L. paracasei* HII01 had any effects on the skeletal muscle glucose transport system, the basal and insulin-stimulated glucose uptakes in the isolated hemi-diaphragm were determined (Appendix A). Our results found that the rate of basal glucose uptake and insulin-stimulated glucose uptake by the hemi-diaphragm in the DMC group significantly decreased compared with the NDC group (*p* < 0.05). Likewise, the insulin-stimulated glucose uptake and the delta glucose uptake, which was calculated as insulin-treated minus basal glucose uptake for paired muscles, in the DMC group were significantly reduced compared with the NDC group (*p* < 0.05). These findings implied an impairment of insulin action in skeletal muscle. In contrast, the administration of *L. paracasei* HII01 for 12 weeks significantly enhanced the rates of insulin-stimulated glucose uptake and delta glucose uptake compared with the DMC group (28.94%, *p* < 0.05). Similarly, significant increases in the rates of insulin-stimulated glucose uptake and delta glucose uptake were precisely noted in the DMM and DMM-L groups compared with the DMC group (50.58% and 43.03%, respectively, *p* < 0.05) (Appendix A). 

### 3.7. Effects of L. paracasei HII01 on Protein Expressions of GLUT4, pAkt^Ser473^, pAMPK^Thr172^, NF-kB and TNF-α in Soleus Muscle

To elucidate the underlying mechanisms sustaining the possible beneficial effects of *L. paracasei* HII01 regarding improvement of insulin-stimulated glucose uptake, the expressions of key proteins involved in insulin-stimulated glucose uptake, such as GLUT4 protein expression and Akt^Ser473^ phosphorylation in soleus muscle, were investigated. In normal rats, supplementation of *L. paracasei* HII01 for 12 weeks did not alter the GLUT4 protein expression compared with the NDC group (Figure 3A). As expected, the expression of GLUT4 protein of the DMC group significantly decreased compared with the NDC group (*p* < 0.01). The protein expressions of GLUT4 were markedly restored in the DM-L, DMM and DMM-L groups compared with the DMC group (*p* < 0.05). As illustrated in Figure 3B, there were no significant differences in pAkt^Ser473^/Akt protein ratio between the NDC and ND-L groups. A reduction in the pAkt^Ser473/^Akt protein ratio was found in the DMC group compared with the NDC group (*p* < 0.05). The administration of *L. paracasei* HII01, metformin alone or in combination with *L. paracasei* HII01 effectively reversed the activation of Akt compared with the DMC group (*p* < 0.05). 

We also evaluated the effects of *L. paracasei* HII01 on AMPK activation. In addition to the insulin signaling proteins, the phosphorylation of AMPK can stimulate GLUT4 translocation for glucose uptake in the skeletal muscle via the insulin-independent pathway. As shown in Figure 4, the oral administration of *L. paracasei* HII01 had no effect on the pAMPK^Thr172^/AMPK protein ratio in normal rats. The DMC group showed a significant decrease in the pAMPK^Thr172^/AMPK protein ratio compared with the NDC group (*p* < 0.05). Interestingly, the administration of probiotic *L. paracasei* HII01, metformin alone or in combination with *L. paracasei* HII01 efficiently recovered the pAMPK ^Thr172^/AMPK protein ratio compared with the DMC group (*p* < 0.05). 

It is well established that in addition to lipid accumulation-induced insulin resistance, chronic inflammation can also induce insulin resistance. Thus, we evaluated the effects of *L. paracasei* HII01 (10^8^ CFU/day) on inflammatory cytokines, NF-kB and TNF-α. As shown in Figure 5A, the NF-kB expression in the DMC group was significantly higher than in the NDC group (*p* < 0.05). However, the oral administration of probiotic *L. paracasei* HII01 successfully reversed that result (*p* < 0.05). However, *L. paracasei* HII01 administration in normal rats did not alter the NF-kB protein expression. The expression of TNF-α is shown in Figure 5B. The administration of probiotic *L. paracasei* HII01 did not affect the TNF-α protein expressions in the normal rats (*p* > 0.05). The protein expression of TNF-α was significantly higher in the DMC group than in the NDC group (*p* < 0.05). Compared with the DMC group, the protein expressions of TNF-α were significantly reduced in the DMM, DM-L and DMM-L groups (*p* < 0.05). 

### 3.8. Effects of L. paracasei HII01 on Plasma Endotoxemia

It is well accepted that endotoxemia, characterized by excess circulating bacterial wall LPS, is associated with systemic inflammation and T2DM. Consequently, we measured the plasma LPS levels. As shown in Figure 6, the DMC group had a significantly higher plasma LPS level than the NDC group (*p* < 0.05). Remarkably, the administration of metformin, *L. paracasei* HII01 alone or in combination with metformin significantly reduced the plasma LPS level compared with the DMC group (*p* < 0.05). 

### 3.9. Effects of L. paracasei HII01 on Intestinal Permeability

Since the underling mechanisms behind the reinforcement of the increased plasma LPS level is expected to involve the gut permeability, we also examined the integrity of the intestinal membrane. This was carried out using an indirect method for the assessment of gut leakiness: measuring the level of DX-4000–FITC in plasma. The plasma levels of DX-4000-FITC of diabetic rats at 2.5 and 5 h were significantly higher than those of normal rats (Figure 7), indicating that an increase in intestinal permeability was found in diabetic rats. Interestingly, treatment with probiotic *L. paracasei* HII01, metformin alone or in combination with *L. paracasei* HII01 significantly reduced the plasma DX-4000-FITC level at 2.5 h compared with the DMC group (*p* < 0.05). In addition, in comparison with the DMC group, treatment with *L. paracasei* HII01 combined with metformin significantly decreased the plasma DX-4000-FITC level at 5 h (*p* < 0.05). 

### 3.10. Effects of L. paracasei HII01 on Short-Chain Fatty Acids in Cecal Content

SCFAs are carbon chain 1–6 organic fatty acids that are generated from the fermentation of undigested starch and fiber by lactic acid bacteria. The major SCFAs are lactic acid, propionic acid, butyric acid and acetic acid. Thus, we evaluated the effects of *L. paracasei* HII01 on the levels of SCFAs in cecal content. As shown in Appendix A, there was no significant difference in lactic acid level in normal rats administered *L. paracasei* HII01 for 12 weeks compared with normal control rats. However, the lactic acid level in the DMC group was significantly reduced compared with the NDC group (*p* < 0.05). Interestingly, treatment with *L. paracasei* HII01, metformin alone or in combination with *L. paracasei* HII01 significantly increased the level of lactic acid compared with the DMC group (*p* < 0.05). Furthermore, the DMM group had significantly increased lactic acid level compared with the DM-L group (*p* < 0.05). The level of propionic acid in normal rats administered *L. paracasei* HII01 was similar to that of normal control rats (*p* > 0.05). The DMC group had a significantly reduced level of propionic acid compared with the NCD group (*p* < 0.05). Meanwhile, the propionic acid levels in all treatment groups were significantly increased compared with the DMC group (*p* < 0.05). Moreover, the combined treatment of *L. paracasei* HII01 and metformin was significantly enhanced compared to the DM-L group (*p* < 0.05). The butyric acid level in the DMC group did not differ from that of the NDC group (*p* < 0.05). However, treatment with *L. paracasei* HII01, metformin alone or in combination with *L. paracasei* HII01 significantly increased the level of butyric acid compared with the DMC group (*p* < 0.05). The acetic acid levels in normal rats administered *L. paracasei* HII01 were significantly higher than those of the NDC rats (*p* < 0.05). In addition, the level of acetic acid in the DMC group did not significantly differ from that of the NDC group (*p* < 0.05). Nevertheless, the administration of probiotic *L. paracasei* HII01 to diabetic rats significantly enhanced the acetic acid level compared with the DMC group (*p* < 0.05). 

### 3.11. Effects of L. paracasei HII01 on the Bacterial DNA in Feces

The relative abundance of Lactobacillus and Bifidobacterium spp. in the gut microbiota is shown in Appendix A (as a Appendix A). In comparison with the NDC group, the number of fecal Lactobacillus spp. was significantly altered in normal rats that received *L. paracasei* HII01 (*p* < 0.05). On the other hand, the DMC group had a significantly decreased number of fecal Lactobacillus spp. compared with the NDC group (*p* < 0.05). Oral administration of *L. paracasei* HII01, metformin alone or in combination with *L. paracasei* HII01 to diabetic rats significantly increased the number of fecal Lactobacillus spp. compared with the DMC rats (*p* < 0.05). However, the DMC group had a higher number of fecal Bifidobacterium spp. than the NDC group (*p* < 0.05). The administration of metformin, *L. paracasei* HII01 alone or in combination with metformin to diabetic rats significantly increased the number of fecal Bifidobacterium spp. compared with the DMC group (*p* < 0.05). Additionally, the number of this beneficial bacteria significantly increased in the DMM and DMM-L groups in comparison with the DM-L group (*p* < 0.01). As illustrated in Appendix A, the result revealed that the numbers of fecal *E. coli* and *C. perfringens* in the ND-L group were significantly lessened compared with the NDC group (*p* < 0.05). The DMC group had a higher number of fecal *E. coli* than the NDC group (*p* < 0.05). Interestingly, all the treatment groups had significantly reduced *E. coli* numbers compared with the DMC group (*p* < 0.05). The number of *C. perfringens*, known as bad bacteria, in the DMC group did not differ from that of the NDC group. However oral administration of metformin, *L. paracasei* HII01 alone or in combination with metformin to diabetic rats significantly reduced the number of Fecal *C. perfringens* compared with the DMC group (*p* < 0.05). Interestingly, the combination of probiotic *L. paracasei* HII01 and metformin significantly decreased the number of *C. perfringens* compared with the diabetic rats treated with probiotic alone (*p* < 0.05). 

## 4. Discussion

The present study was undertaken to investigate the possible beneficial effect of *L. paracasei* HII01 on glycemia in type 2 diabetic rat model. Our results demonstrated that the administration of *L. paracasei* HII01 at a dose of 10^8^ CFU/day for 12 weeks effectively resulted in the following: (1) reduction in fasting plasma glucose, insulin, leptin and lipids levels as well as improvement in glucose intolerance; (2) improvement of PI3K/Akt signaling and AMPK activation, which are involved in enhancing the rate of insulin-stimulated glucose uptake of the isolated hemi-diaphragm; (3) modulation of gut microbiota and subsequent amelioration of plasma endotoxemia. 

The type 2 diabetic rat model used in this study presented the general characteristics of T2DM, including obesity, hyperglycemia, insulin resistance, impaired glucose tolerance and dyslipidemia, similar to T2DM patients [28]. In addition, the plasma LPS level was significantly increased, which, at least in part, is linked to insulin resistance in untreated diabetic rats. Similar to humans, our findings also found that an abundance of pathogenic bacteria, *E. coli* and *C. perfringens* in diabetic rats [29,30]. At the end of the study, we found that *L. paracasei* HII01 administration significantly improved not only BW, VF/BW, and plasma lipid levels (TG, cholesterol, LDL, and HDL) but also reduced the fasting blood glucose (FBG) levels and improved glucose tolerance, demonstrating its antidiabetic effect. Nevertheless, the beneficial effect of *L. paracasei* HII01 on glycemic control in the present study is not linked to its insulinotropic action. The anti-hyperglycemic effect of *L. paracasei* HII01 might be explained by other mechanisms, such as enhanced insulin sensitivity or relevant glucose uptake, the same as the results found in the metformin treatment group. This assertion is supported by the HOMA-IR index and the outcomes of OGTT. Additionally, the results of this study revealed that gut dysbiosis was attenuated after 12 weeks of oral administration of *L. paracasei* HII01 in diabetic rats. It was interesting to note that the amelioration efficiency of *L. paracasei* HII01 on those blood metabolic parameters were close to the results of studies involving diabetic rats treated with metformin alone or combination with *L. paracasei* HII01. It was suggested that there were no synergistic or additive effects of metformin and probiotic *L. paracasei* HII01, particularly induction of hypoglycemia.

Alteration of adipokine is one of the possible mechanisms contributing to hyperglycemia and insulin resistance of diabetes. This study demonstrated that *L. paracasei* HII01 supplement effectively modulated adipokine imbalance in diabetic rats. Adiponectin acts as an insulin-sensitizer, which serves to enhance fatty acid oxidation, reduce tissue TG accumulation and, finally, improve insulin signaling [31]. While, leptin is a hormone that is important for glucose homeostasis by stimulating the PI3K signaling pathway [32]. The current study has demonstrated that the decreased leptin level is related to an improvement in insulin sensitivity and reduction in plasma glucose and lipid levels in type 2 diabetic rats treated with *L. casei* CCFM419 [33]. The reduction in visceral fat accumulation acknowledged in the present study might be involved with the suppression in plasma leptin and increasing in plasma adiponectin levels, which contribute to the improvement in insulin sensitivity, anti-hyperglycemia as well as anti-hyperlipidemia in diabetic rats treated with *L. paracasei* HII01. However, there are several mechanisms involved to the pathogenesis of insulin resistance. Among these, genes involved in adipose tissue metabolism can be considered possibly responsible for insulin sensitivity. Petrone A et al., 2007, reported that the promoter region of the adiponectin gene (+45T>G Adiponectin SNPs) could influence adiponectin levels and, consequently, insulin sensitivity in obesity and diabetes mellitus. [34].

Next, to explore more about the molecular mechanisms of *L. paracasei* HII01 in insulin sensitivity, we further evaluated the insulin signaling and glucose transport system in the skeletal muscle since the skeletal muscle is the major site of glucose uptake in the postprandial state in normal condition. Comparable with other studies, the results demonstrated that the insulin action is diminished and the ability of insulin to stimulate glucose uptake is blunted in type 2 diabetic condition [35]. Although the mechanisms of insulin resistance are not fully understood, lipid and inflammation-induced insulin resistance is one of the potential candidate mechanisms for insulin resistance [36]. Previous studies found that the TLR-4-LPS pathway can stimulate pro-inflammatory cytokines in the skeletal muscle, and pro-inflammatory cytokines, such as TNF-α, promote insulin resistance though NF-kB inflammatory signaling [37]. In addition, the rate of insulin-stimulated glucose uptake depends on the phosphorylation of Akt, the major insulin signaling protein, and the total or membrane GLUT4 protein expression [38]. Interestingly, we found that *L. paracasei* HII01 effectively increased Akt^Ser473^ phosphorylation and GLUT4 protein expression as well as decreased the expression of TNF-α and NF-kB in the skeletal muscle. Similarly, the supplementation of a combination of the probiotics *L. rhamnosus*, *L. acidophilus* and *B. bifidium* enhanced pAkt^Ser473^ in the muscle of diet-induced obese (DIO) mice [39]. Besides, in TNF-α treated L6 cells, probiotic *B. lactis* HY8101 treatment increased insulin-stimulated phosphorylation of Akt^Ser473^ and GLUT4 protein [40]. Cumulatively, our data suggested that *L. paracasei* HII01 treatment improved the insulin signaling pathway, at least partly, through the reduction in systemic inflammation. Furthermore, the activation of AMPK, an energy-sensing enzyme, is one of the possible mechanisms linked to glucose uptake via directly activating GLUT4 translocation to membrane in skeletal muscles [41]. Metformin, a first-line drug for T2DM treatment, exerts its action mainly by activating AMPK. We also found that *L. paracasei* HII01 supplementation enhanced AMPK^Thr172^ phosphorylation in the skeletal muscle of diabetic rats; this was also the case in the metformin treatment group. 

Lactic, propionic, butyric and acetic acids are the most important SCFAs that affect glycemic control [42]. Importantly, the results demonstrated that the administration of *L. paracasei* HII01 for 12 weeks effectively enhanced the number of those SCFAs in cecal content. It is well known that the receptors of SCFAs are two G-protein coupled receptors (GPCRs), free fatty acid receptor 2 (FFAR2) and FFAR3, which are widely expressed in the skeletal muscle, intestinal, adipose, liver and pancreatic tissues [43]. In addition, these SCFAs are not only of importance in gut health and as signaling molecules but might also enter the systemic circulation and directly affect metabolism or the function of peripheral tissues via the AMPK activation [44]. Thus, SCFAs may partly influence glucose metabolism and insulin resistance in type 2 diabetic rats supplemented with *L. paracasei* HII01. 

Recent studies reported that changes in gut microbiota composition are associated with an increase in gut LPS. The TLR4–LPS complex triggers the pro-inflammatory cytokines, which cause intestinal inflammation and decrease in tight junction proteins [45]. The loss of tight junction proteins is linked to increased gut permeability and the subsequent leakage of LPS to systemic circulation [46]. Interestingly, these abnormalities were attenuated after our probiotic *L. paracasei* HII01 administration for 12 weeks. However, the mechanisms were not investigated in this study. Lim et al., 2016, found that the supplementation of *L. sakei* OK67 suppressed TLR-4 expression and the NF-kB pathway, which are involved in the reduction in the level of intestinal TNF-α and interlukin-6 (IL-6) and, subsequently, the increase in tight junction protein, including zonula occludens (ZO-1), occludin and claudin expression in the colon of obese mice [15]. We further hypothesize that the gut microbiota modulation and anti-inflammatory effects of *L. paracasei* HII01 could be another probable underlying mechanism for the improvement in gut permeability and, subsequently, endotoxemia.

There are limitations to this study that have to be considered. Firstly, we showed the protective effect of *L. paracasei* HII01 administration on gut barrier integrity by measuring the concentration of LPS in the circulation and plasma levels of DX-4000-FITC, which is an indirect method for the assessment of gut leakiness. A direct assessment of intestinal tight junction permeability or levels of intestinal tight junction markers, such as occludin or ZO-1, would better confirm the role of our probiotic in preserving the intestinal epithelial barrier. Lastly, we used an experimental diabetic rat model to test our hypothesis and the results cannot be directly extrapolated to humans due to differences in gut microbiota and physiology.

## 5. Conclusions

This study provided the first evidence that *L. paracasei* HII01 administration ameliorates hyperglycemia and enhances insulin stimulated glucose uptake in HFD–STZ induced type 2 diabetic rats. These effects are associated with modulation of gut microbiota along with gut permeability, leading to improved systemic endotoxemia and inflammation-disturbed insulin sensitivity in the skeletal muscle through PI3K/Akt signaling and AMPK activation as summarized in Figure 8. Thus, *L. paracasei* HII01 has the potential for development as a complementary supplement strategy for type 2 diabetic patients. 

## Figures and Tables

**Figure 1 nutrients-12-03015-f001:**
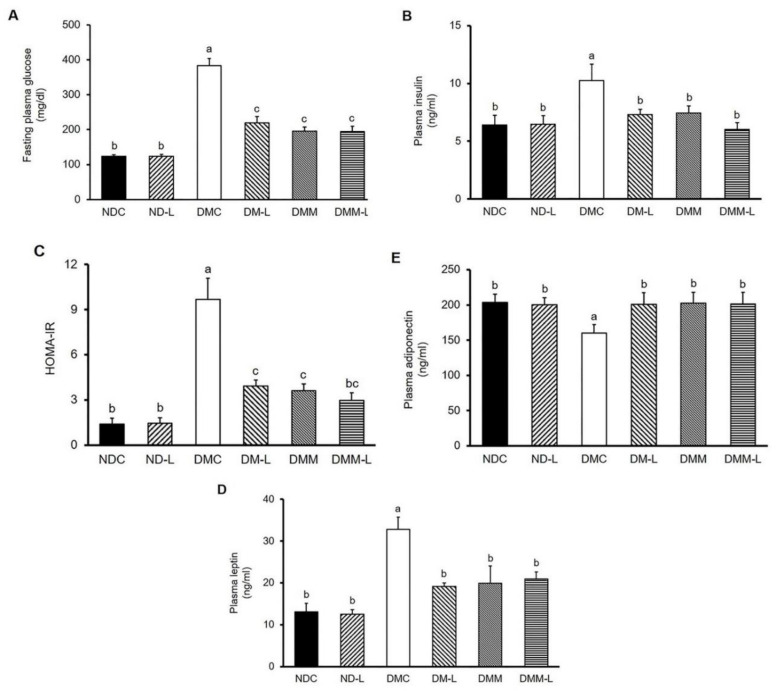
Effects of *L. paracasei* HII01 on the fasting plasma levels of (**A**) glucose, (**B**) insulin, (**C**) HOMA-IR, (**D**) leptin and (**E**) adiponectin in experimental rats. NDC, normal control rats; ND-L, normal control rats supplemented with *L. paracasei* HII01 (10^8^ CFU/day); DMC, diabetic rats control; DM-L, diabetic rats supplemented with *L. paracasei* HII01 (10^8^ CFU/day); DMM, diabetic rats treated with metformin 30 mg/kg; DMM-L, diabetic rats supplemented with combination of *L. paracasei* HII01 (10^8^ CFU/day) and metformin 30 mg/kg; HOMA-IR index, homeostasis model assessment of insulin resistance. All data are expressed as mean ± SEM. Different lowercase letters indicate significant differences among different groups (*p* < 0.05).

**Figure 2 nutrients-12-03015-f002:**
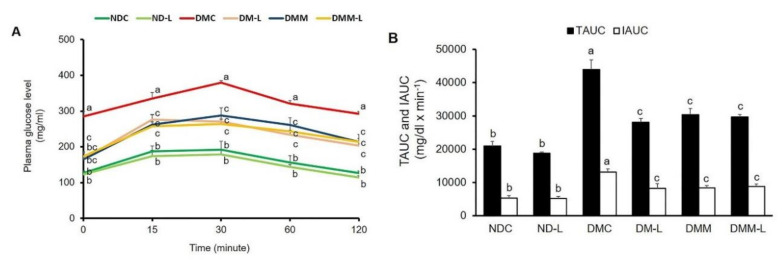
Effects of *L. paracasei* HII01 on the OGTT in experimental group. (**A**) Glucose response; (**B**) area under the curve for glucose. NDC, normal control rats; ND-L, normal control rats supplemented with *L. paracasei* HII01 (10^8^ CFU/day); DMC, diabetic rats control; DM-L, diabetic rats supplemented with *L. paracasei* HII01 (10^8^ CFU/day); DMM, diabetic rats treated with metformin 30 mg/kg; DMM-L, diabetic rats supplemented with combination of *L. paracasei* HII01 (10^8^ CFU/day) and metformin 30 mg/kg; OGTT, oral glucose tolerance test; TAUC, total area under the curve; IAUC, incremental area under the curve. All data are expressed as mean ± SEM. Different lowercase letters indicate significant differences among different groups (*p* < 0.05).

**Figure 3 nutrients-12-03015-f003:**
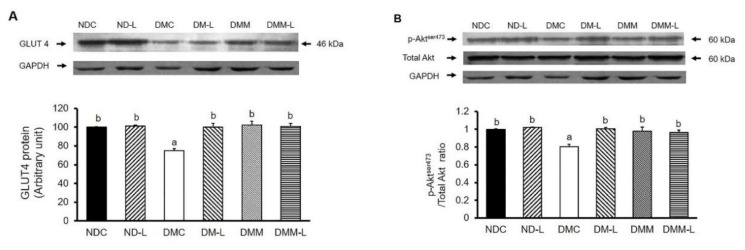
Effects of *L. paracasei* HII01 on Western blotting of insulin-stimulated glucose uptake marker proteins (**A**) GLUT4 protein (**B**) pAkt^ser473^/Total Akt ratio in experimental groups. NDC, normal control rats; ND-L, normal control rats supplemented with *L. paracasei* HII01 (10^8^ CFU/day); DMC, diabetic rats control; DM-L, diabetic rats supplemented with *L. paracasei* HII01 (10^8^ CFU/day); DMM, diabetic rats treated with metformin 30 mg/kg; DMM-L, diabetic rats supplemented with combination of *L. paracasei* HII01 (10^8^ CFU/day) and metformin 30 mg/kg. GLUT4, glucose transporter 4; pAkt^Ser473^/total Akt ratio, phosphorylation of protein kinase B per total protein kinase B ratio. All data are expressed as mean ± SEM. Different lowercase letters indicate significant differences among different groups (*p* < 0.05).

**Figure 4 nutrients-12-03015-f004:**
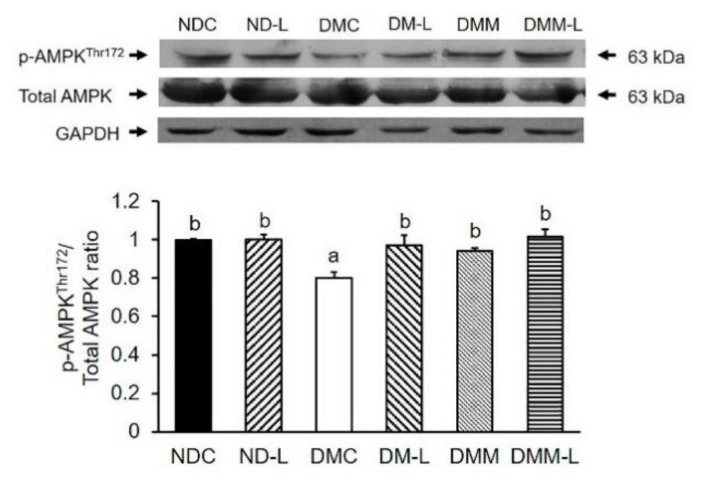
Effects of *L. paracasei* HII01 on Western blotting of pAMPK^Thr172^/Total AMPK ratio in skeletal muscle of experimental groups. NDC, normal control rats; ND-L, normal control rats supplemented with *L. paracasei* HII01 (10^8^ CFU/day); DMC, diabetic rats control; DM-L, diabetic rats supplemented with *L. paracasei* HII01 (10^8^ CFU/day); DMM, diabetic rats treated with metformin 30 mg/kg; DMM-L, diabetic rats supplemented with combination of *L. paracasei* HII01 (10^8^ CFU/day) and metformin 30 mg/kg. pAMPK^Thr172^/ total AMPK ratio, phosphorylation of AMP-activated protein kinase per total AMPK ratio. All data are expressed as mean ± SEM. Different lowercase letters indicate significant differences among different groups (*p* < 0.05).

**Figure 5 nutrients-12-03015-f005:**
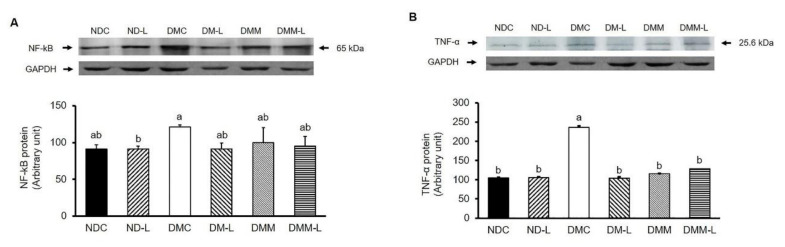
Effects of *L. paracasei* HII01 on Western blotting of inflammatory protein marker (**A**) NF-kB (**B**) TNF-α in skeletal muscle of experimental groups. NDC, normal control rats; ND-L, normal control rats supplemented with *L. paracasei* HII01 (10^8^ CFU/day); DMC, diabetic rats control; DM-L, diabetic rats supplemented with *L. paracasei* HII01 (10^8^ CFU/day); DMM, diabetic rats treated with metformin 30 mg/kg; DMM-L, diabetic rats supplemented with combination of *L. paracasei* HII01 (10^8^ CFU/day) and metformin 30 mg/kg. NF-kB, nuclear factor-kappa B; TNF-α, tumor necrosis factor alpha. All data are expressed as mean ± SEM. Different lowercase letters indicate significant differences among different groups (*p* < 0.05).

**Figure 6 nutrients-12-03015-f006:**
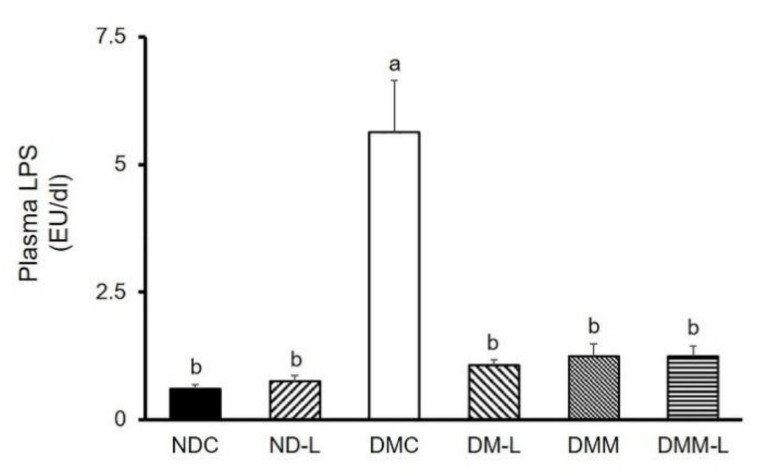
Effects of *L. paracasei* HII01 on the plasma LPS levels in experimental groups. NDC, normal control rats; ND-L, normal control rats supplemented with *L. paracasei* HII01 (10^8^ CFU/day); DMC, diabetic rats control; DM-L, diabetic rats supplemented with *L. paracasei* HII01 (10^8^ CFU/day); DMM, diabetic rats treated with metformin 30 mg/kg; DMM-L, diabetic rats supplemented with combination of *L. paracasei* HII01 (10^8^ CFU/day) and metformin 30 mg/kg. LPS, lipopolysaccharide. All data are expressed as mean ± SEM. Different lowercase letters indicate significant differences among different groups (*p* < 0.05).

**Figure 7 nutrients-12-03015-f007:**
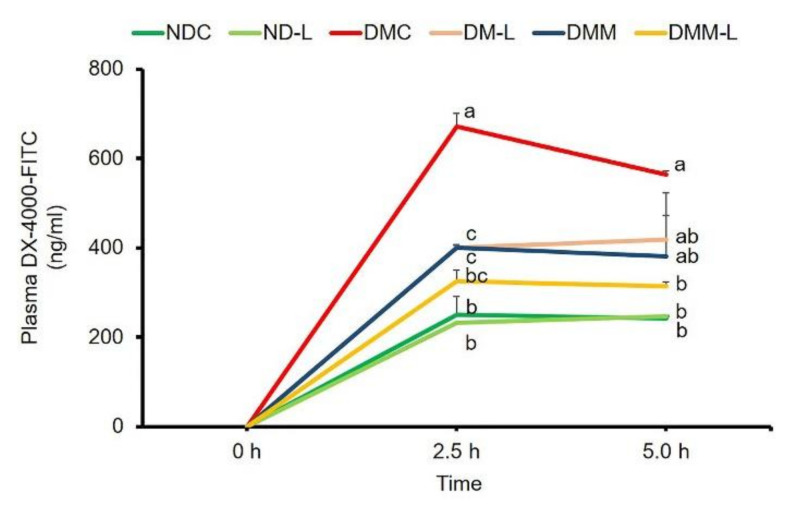
Effects of *L. paracasei* HII01 on gut permeability measured by plasma FITC-fluorescent dye levels in experimental groups. NDC, normal control rats; ND-L, normal control rats supplemented with *L. paracasei* HII01 (10^8^ CFU/day); DMC, diabetic rats control; DM-L, diabetic rats supplemented with *L. paracasei* HII01 (10^8^ CFU/day); DMM, diabetic rats treated with metformin 30 mg/kg; DMM-L, diabetic rats supplemented with combination of *L. paracasei* HII01 (10^8^ CFU/day) and metformin 30 mg/kg; 4000-DX-FITC, 4000 Da Fluorescein isothiocyanate–dextran. All data are expressed as mean ± SEM. Different lowercase letters indicate significant differences among different groups (*p* < 0.05).

**Figure 8 nutrients-12-03015-f008:**
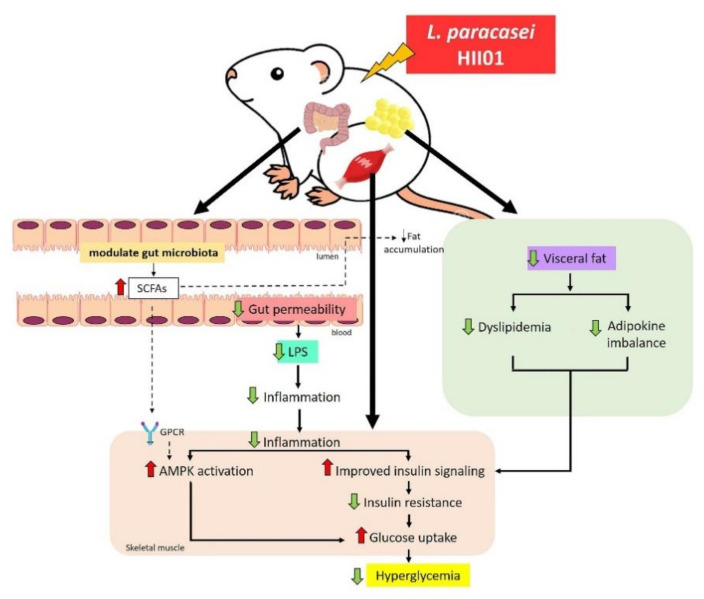
Possible mechanism of *L. paracasei* HII01 in type 2 diabetic rats.

**Table 1 nutrients-12-03015-t001:** Effects of *L. paracasei* HII01 on BW, VF weight and VF/100 g BW in experimental groups.

Parameters	NDC	ND-L	DMC	DM-L	DMM	DMM-L
BW (g)	553.33 ± 18.60 ^b^	568.33 ± 26.13 ^b^	682.50 ± 35.25 ^a^	575.00 ± 13.78 ^b^	586.25 ± 11.43 ^b^	589.00 ± 4.00 ^b^
VF (g)	45.17 ± 2.36 ^b^	41.00 ± 3.20 ^b^	86.33 ± 7.84 ^a^	62.8 ± 2.99 ^c^	61.75 ± 2.46 ^c^	60.40 ± 5.62 ^c^
VF/100g BW	8.13 ± 0.22 ^b^	7.23 ± 0.53 ^b^	12.62 ± 0.51 ^a^	10.89 ± 0.27 ^c^	10.54 ± 0.41 ^c^	10.27 ± 1.00 ^c^

NDC, normal control rats; ND-L, normal control rats supplemented with *L. paracasei* HII01 (10^8^ CFU/day); DMC, diabetic rats control; DM-L, diabetic rats supplemented with *L. paracasei* HII01 (10^8^ CFU/day); DMM, diabetic rats treated with metformin 30 mg/kg; DMM-L, diabetic rats supplemented with combination of *L. paracasei* HII01 (10^8^ CFU/day) and metformin 30 mg/kg; BW, body weight; VF, visceral fat. All data are expressed as mean ± SEM. Different lowercase letters indicate significant differences among different groups (*p* < 0.05).

**Table 2 nutrients-12-03015-t002:** Effects of *L. paracasei* HII01 on lipid parameters in experimental groups.

Parameters (mg/dL)	NDC	ND-L	DMC	DM-L	DMM	DMM-L
Triglyceride	38.35 ± 1.26 ^b^	31.82 ± 1.67 ^b^	83.57 ± 7.18 ^a^	35.76 ± 2.78 ^b^	33.88 ± 1.02 ^b^	38.15 ± 3.55 ^b^
Cholesterol	44.35 ± 1.04 ^b^	40.53 ± 2.48 ^b^	72.92 ± 6.02 ^a^	42.85 ± 1.81^b^	35.13 ± 2.63 ^c^	33.85 ± 3.21^c^
HDL	59.75 ± 1.55 ^b^	64.00 ± 3.67 ^ab^	69.00 ± 5.98 ^ab^	72.33 ± 1.11^a^	71.00 ± 2.64 ^a^	72.33 ± 0.76 ^a^
LDL	10.00 ± 1.22 ^b^	10.08 ± 1.32 ^b^	23.75 ± 3.50 ^a^	16.67 ± 3.28 ^c^	14.00 ± 1.30 ^b^	13.00 ± 1.14 ^b^

**N**DC, normal control rats; ND-L, normal control rats supplemented with *L. paracasei* HII01 (10^8^ CFU/day); DMC, diabetic rats control; DM-L, diabetic rats supplemented with *L. paracasei* HII01 (10^8^ CFU/day); DMM, diabetic rats treated with metformin 30 mg/kg; DMM-L, diabetic rats supplemented with combination of *L. paracasei* HII01 (10^8^ CFU/day) and metformin 30 mg/kg; HDL, high-density lipoprotein; LDL, low-density lipoprotein. All data are expressed as mean ± SEM. Different lowercase letters indicate significant differences among different groups (*p* < 0.05).

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
