# Peer review of "Putative Mechanisms Responsible for the Antihyperglycemic Action of Lactobacillus paracasei HII01 in Experimental Type 2 Diabetic Rats"

_nutrients, 2020, doi:10.3390/nu12103015_

Round 1

Reviewer 1 Report

In the present study, the authors evaluated the antihyperglycemic efficacy of Lactobacillus paracasei HII01 and its possible mechanisms of action in diabetic rats. L. paracasei HII01 improved glycemic parameters, glucose uptake, insulin-signaling proteins including pAktSer473, GLUT4 and pAMPKThr172, TNF-α and NF-kB  in diabetic rats. The authors concluded that  L. paracasei  HII01 amelioreted hyperglycemia in diabetic rats by modulating gut microbiota along with  lessening leaky gut, leading to improvement in endotoxemia and inflammation-disturbed insulin  signaling, which was mediated partly by PI3K/Akt signaling and AMPK activation.  This is an interesting study that can add new insight in the treatment of experimental type 2 diabetes.

The Authors have to address some comments/questions trying to be focused on the results of this study. 

Major points

  1. As the Authors underlined in the introduction section, T2D is a multifactorial disease where genetics, lifestyle, overweight/obesity, all play a relevant role. In this regard also adiponectin and leptin levels may be determined or influenced by gene polymorphisms (Petrone A et al Int J. of Obes. 2006), please comment on that, referring to this study on the discussion section.
  2. High sugar and high fat impairs cardiac function in animal experimental model (Carbone et al Int. J of Cardiology, 2015), did you observe any beneficial effect in your experimental model on this concern? Please comment on that.
  3. The hypothesis concerning the reduction of obesity is purely speculative and should be removed by Fig.11 that has to report only hypotheses directly linked to the result of the present study
  4. Table 3 and fig. 3, 9 and 10 should be transfer in supplementary materials
  5. The discussion should be shortened and very focused only on the results of the present study avoiding speculation on data not reported

Author Response

We wish to express our appreciation to the Reviewers for their insightful comments, which have helped us significantly to improve our manuscript. According to the suggestions, we have thoroughly revised our manuscript. The revised manuscript file was indicated it with yellow highlight for added or rewritten contents, please see a revised version. Point-by-point responses to the comments are listed below.

Point 1. As the Authors underlined in the introduction section, T2D is a multifactorial disease where genetics, lifestyle, overweight/obesity, all play a relevant role. In this regard also adiponectin and leptin levels may be determined or influenced by gene polymorphisms (Petrone A et al Int J. of Obes. 2006), please comment on that, referring to this study on the discussion section.

Response 1: We appreciate your comment very much. According to study of Petrone A et al. 2006, the influence of promoter region of the adiponectin gene (+45T>G Adiponectin SNPs) on adiponectin levels and, consequently, insulin sensitivity in obesity and diabetes mellitus was add to the discussion section (Paragraph 2, Line 545-549). However, our study uses a model of high fat diet combined with streptozotocin induced type 2 diabetes. Therefore, the genetic factor may not be considered mainly responsible for insulin sensitivity.

Point 2. High sugar and high fat impairs cardiac function in animal experimental model (Carbone et al Int. J of Cardiology, 2015), did you observe any beneficial effect in your experimental model on this concern? Please comment on that.

Response 2: We deeply appreciate the reviewer's comment on this point. Although, several studies have demonstrated the cardiac dysfunction in experimental diabetic rat models. The main objective of our study was to evaluate the effect of L. paracasei HII01 on glycemic control in experimental rats. Therefore, the cardiac functions and blood pressure were not performed.

Point 3. The hypothesis concerning the reduction of obesity is purely speculative and should be removed by Fig.11 that has to report only hypotheses directly linked to the result of the present study.

Response 3: We profoundly appreciate the reviewer's comment on this point. The figure 8 (page 17, line 615); Possible mechanism of L. paracasei HII01 in type 2 diabetic rats was corrected. The term of “reduction of obesity” was removed. However, the reduction of visceral fat might involve to the modulation of adipokine imbalance in all treatment groups.

Point 4. Table 3 and fig. 3, 9 and 10 should be transfer in supplementary materials.

Response 4: In accordance with the reviewer's comment, we have transfer Table 3, Figure 3, 9, 10 to the supplementary files as a Table S3, Figure S1, Figure S2 and Figure S3, respectively.

Point 5. The discussion should be shortened and very focused only on the results of the present study avoiding speculation on data not reported.

Response 5: In accordance with the reviewer's comment, we have reconstructed our discussion in order to be more shortened and described via our findings.

Yours sincerely,

Narissara Lailerd, PhD.

Reviewer 2 Report

The manuscript nutrients-937573 titled “Putative mechanisms responsible for the antihyperglycemic action of Lactobacillus paracasei HII01 in experimental Type 2 diabetic rats” by Toejing et al. have reported beneficial effects using a probiotic strain by modulating gut microbiota along with lessening leaky gut, leading to improvement in endotoxemia and inflammation-disturbed insulin signaling, which was mediated partly by PI3K/Akt signaling and AMPK activation. I have a few concerns regarding the present manuscript.

-The manuscript is a well-structured study, thanks to the authors for that. There are some typographic errors regarding the correct use of italics in the strain names, please check
-Also check the abbreviations, only explain the first time
-I read with interest the present work and I like the way in express the obtained results, however differences were especially in DMC groups, what is the reason do not observe changes in the other groups, the authors have any idea?
-Statistical analysis is too short, it is possible to explain in more words this section?
-Limitations are missing in this manuscript, the authors have found one in the present study

Author Response

We wish to express our appreciation to the Reviewers for their insightful comments, which have helped us significantly to improve our manuscript. According to the suggestions, we have thoroughly revised our manuscript. The revised manuscript file was indicated it with blue highlight for added or rewritten contents, please see a revised version. Point-by-point responses to the comments are listed below.

Point 1. The manuscript is a well-structured study, thanks to the authors for that. There are some typographic errors regarding the correct use of italics in the strain names, please check.

Response 1: In accordance with the reviewer's comment, we have rechecked and corrected those typing errors.

Point 2. Also check the abbreviations, only explain the first time.

Response 2: In accordance with the reviewer's comment, we have rechecked and corrected those typing errors.

Point 3. I read with interest the present work and I like the way in express the obtained results, however differences were especially in DMC groups, what is the reason do not observe changes in the other groups, the authors have any idea?

Response 3: We deeply appreciate the reviewer's comment on this point. The results from our study showed that the effects of administration of metformin, L. paracasei HII01 alone or in combination with metformin in diabetic rats had the similar efficiency

on glycemic control and modulation of gut dysbiosis. In my opinion, these findings suggested that there were no synergistic or additive effects of metformin and probiotic L. paracasei HII01. So, consumption of probiotic L. paracasei HII01 may provide the health beneficial effects in diabetic patients such as improvement of inflammatory status without hypoglycemia. Also, I have added this interpretation in Discussion part (page 14 of 19, paragraph 2, line 531-534).

Point 4. Statistical analysis is too short, it is possible to explain in more words this section?

Response 4: We are grateful for your comment. We have given more explanations in the part of Statistical analysis on page 5 of 19, line 216-222.

Point 5. Limitations are missing in this manuscript, the authors have found one in the present study

Response 5: We appreciate your comment very much to this point. We have added the limitations of this present study on page 16 of 19, paragraph 3 and line 597-604.

Yours sincerely,

Narissara Lailerd, PhD.

Round 2

Reviewer 1 Report

the Authors have answered all the questions raised